# A single-dose mRNA vaccine provides a long-term protection for hACE2 transgenic mice from SARS-CoV-2

Qingrui Huang [1,8], Kai Ji [1,2,8], Siyu Tian [1,2,8], Fengze Wang [1,2,8], Baoying Huang[3], Zhou Tong[4,5], Shuguang Tan[4], Junfeng Hao [6], Qihui Wang [1,4], Wenjie Tan [3✉], George F. Gao [2,4,7✉] & Jinghua Yan [1,2,4✉]

The rapid expansion of the COVID-19 pandemic has made the development of a SARS-CoV-2 vaccine a global health and economic priority. Taking advantage of versatility and rapid development, three SARS-CoV-2 mRNA vaccine candidates have entered clinical trials with a two-dose immunization regimen. However, the waning antibody response in convalescent patients after SARS-CoV-2 infection and the emergence of human re-infection have raised widespread concerns about a possible short duration of SARS-CoV-2 vaccine protection. Here, we developed a nucleoside-modified mRNA vaccine in lipid-encapsulated form that encoded the SARS-CoV-2 RBD, termed as mRNA-RBD. A single immunization of mRNA-RBD elicited both robust neutralizing antibody and cellular responses, and conferred a near-complete protection against wild SARS-CoV-2 infection in the lungs of hACE2 transgenic mice. Noticeably, the high levels of neutralizing antibodies in BALB/c mice induced by mRNA-RBD vaccination were maintained for at least 6.5 months and conferred a long-term notable protection for hACE2 transgenic mice against SARS-CoV-2 infection in a sera transfer study. These data demonstrated that a single dose of mRNA-RBD provided long-term protection against SARS-CoV-2 challenge.

[1] CAS Key Laboratory of Microbial Physiological and Metabolic Engineering, Institute of Microbiology, Chinese Academy of Sciences, Beijing, China. [2] University of Chinese Academy of Sciences, Beijing, China. [3] MHC Key Laboratory of Biosafety, National Institute for Viral Disease Control and Prevention, China CDC, Beijing, China. [4] CAS Key Laboratory of Pathogenic Microbiology and Immunology, Institute of Microbiology, Chinese Academy of Sciences, Beijing, China. [5] Shanxi Academy of Advanced Research and Innovation, Taiyuan, China. [6] Core Facility for Protein Research, Institute of Biophysics, Chinese Academy of Sciences, Beijing, China. [7] Chinese Center for Disease Control and Prevention (China CDC), Beijing, China. [8] These authors contributed equally: Qingrui Huang, Kai Ji, Siyu Tian, Fengze Wang. ✉email: tanwj@ivdc.chinacdc.cn; gaof@im.ac.cn; yanjh@im.ac.cn

Coronavirus disease-19 (COVID-19) caused by SARS-CoV-2 has emerged as a severe global pandemic[1–6]. Since SARS-CoV-2 transmits efficiently from person to person, as of 25 August 2020 more than 23,000,000 cases and 800,000 deaths had been confirmed in 216 countries and territories worldwide. COVID-19 has symptoms ranging from mild disease to severe lung injury and multi-organ failure, eventually leading to death, especially in older patients with other co-morbidities[7–9]. The COVID-19 pandemic has led to not only the enormous burden of mortality and morbidity associated with SARS-CoV-2 infection but also a global economic crisis due to the economic and societal lockdown efforts made in an attempt to thwart the progression of the disease. At present there are no available prophylactics or therapeutics against SARS-CoV-2 infection, highlighting the desperate need for a safe and effective vaccine to halt the ongoing pandemic and prevent new potential outbreaks.

SARS-CoV-2 belongs to the genus *Betacoronavirus* of the family Coronavirdae[10]. Like other human coronaviruses, SARS-CoV-2 surface spike glycoprotein (S) can be cleaved into S1 and S2 subdomains, where the receptor-binding domain (RBD), located at the C-terminal of the S1 subdomain, engages human angiotensin-converting enzyme 2 (hACE2) as the receptor, and S2 mediates membrane fusion. Both the full-length S protein and the RBD are capable of inducing highly potent neutralizing antibodies and cellular immunity[11–14]. Therefore, they have been widely selected as antigens for SARS-CoV-2 vaccine development[12,15–20]. As the vaccine antigen, the RBD can focus the immune response on interference of receptor binding and theoretically entails lower risk of inducing antibodies that readily mediate antibody-dependent enhancement of infection (ADE) compared with the full-length S protein[16,21,22]. A number of highly potent monoclonal antibodies have also been isolated that predominantly target the RBD[23–28]. The crystal structures of the SARS-CoV-2 RBD in complex with hACE2 have been determined by our and other groups[29–31], and this knowledge has further improved our understanding of this vaccine antigen. Therefore, the RBD represents an ideal target for SARS-CoV-2 vaccine development.

An mRNA vaccine has the advantages of safety, rapid development, and potent immunogenicity, especially in lipid-encapsulated form, and multiple mRNA vaccine candidates against infectious diseases or cancer are under clinical development[32]. To date, three SARS-CoV-2 mRNA vaccine candidates that have already advanced to clinical trials are mRNA-1273 encoding the viral S protein from Moderna (the United States of America)[1,33], BNT162b1 expressing the RBD protein from BioNTech (Germany)[34], and ARCoV encoding the RBD protein developed by Abogen (China)[21]. Moreover, mRNA-1273 advanced to Phase III testing on 27 July 2020 and remains one of the leading vaccine candidates against SARS-CoV-2. All three clinical mRNA vaccines are applied in a two-dose immunization regimen[21,33,34]. Recently, a single nucleoside-modified mRNA vaccination has been shown to elicit strong cellular and humoral immune responses against SARS-CoV-2 (ref. [35]). However, protection by a single mRNA vaccination against wild SARS-CoV-2 in animal models is little investigated.

The duration of the neutralizing antibody (NAb) response in humans following coronavirus infection is vital for protection from re-infection. Whereas sustained IgG or NAb levels in individuals were usually maintained for more than 2 years against severe acute respiratory syndrome coronavirus (SARS-CoV) or Middle Eastern respiratory syndrome-related coronavirus (MERS-CoV) infection[36–38], according to recent studies, the NAb response in a high proportion of humans who recovered from SARS-CoV-2 infection only lasted for 2–3 months after infection[39–41]. The transient NAb response following SARS-CoV-2 infection is more similar to the immune responses to endemic seasonal coronaviruses (those associated with the common cold) that have also been reported to be transient[42]. In addition, disease severity enhances the magnitude of the NAb response against SARS-CoV-2 infection but does not affect the kinetics of the NAb response[39]. For those patients with severe symptoms following SARS-CoV-2 infection, declines in NAb titers ranging from 2- to 23-fold over an 18–65-day period have been reported[39]. Noticeably, the first case of human re-infection only a few months after recovery from the first infection has recently been documented[43]. The waning NAb response after SARS-CoV-2 infection has aroused widespread concern over a possible short duration of vaccine protection. Due to the recent emergence of SARS-CoV-2 in the human population, there is currently a paucity of information on the longevity of protection induced by vaccines. Here, we evaluated the protective efficacy of a single RBD-encoding mRNA immunization against wild SARS-CoV-2, investigated the kinetics of humoral response during 6.5 months following vaccination, and we demonstrated the long-term protection of immune sera for hACE2 transgenic mice from SARS-CoV-2 challenge.

## Results

### Construction and characterization of a SARS-CoV-2 mRNA-RBD vaccine.
We designed an mRNA vaccine that encodes the RBD glycoprotein of SARS-CoV-2 strain HB-01 (Fig. 1a) and contains the modified nucleoside $N_1$-methylpseudouridine to prevent innate immune sensing and increase mRNA translation in vivo[44]. Then the expression profiles of the SARS-CoV-2 RBD-encoding mRNA (mRNA-RBD) were characterized by transfecting HEK293T cells. The results demonstrated that the RBD was produced effectively and secreted into the supernatant of HEK293T cells (Fig. 1b). To mediate efficient and prolonged antigen expression by mRNA in vivo, mRNA-RBD was further prepared for vaccination by encapsulation in lipid nanoparticles (LNPs)[44]. Encapsulation efficacy of mRNA-RBD was greater than 92%, based on a ribogreen fluorescence assay. The dynamic light scattering of LNPs in phosphate-buffered saline (PBS) indicated that the average particle size was 78 nm (Fig. 1c), with a narrow PDI of 0.117. Cryo-electron microscopy analysis showed that mRNA-RBD-encapsulated LNPs exhibited an electron-dense core (Fig. 1d). Zeta potential measurement showed that ionization of the surface charge was observed, as the zeta potential increased from −5.8 mV at pH 7.4 to +12.7 mV at pH 4.0 (Fig. 1e). These data suggested that LNPs could respond to pH changes, disrupt the endosomal membranes under an acidic pH, and release mRNA into the cytoplasm.

To examine the duration and distribution of RBD production from mRNA-RBD LNPs in vivo, groups of BALB/c mice ($n = 16$) were immunized with 15 μg mRNA-RBD LNPs via the intramuscular (i.m.) route, and polycytidylic acid (poly(C)) RNA encapsulated in LNPs was used as a placebo control. Four mice per group were euthanized at 6, 12, 24, and 48 h post injection, and serum, muscle, liver, kidney, spleen, and lung samples were obtained for detection for RBD expression level by enzyme-linked immunosorbent assay (ELISA). The results showed that in the mRNA-RBD group, the muscle and liver were the main RBD-expressing tissues, which is in accordance with previous observations for luciferase mRNA LNPs[45], and a very low level of RBD expression was also detected in serum and other tissue samples (Supplementary Fig. 1). Moreover, RBD expression in both muscle and liver tissues peaked at 6 h post inoculation, with average concentrations of 553 and 437 ng/ml and abated relatively quickly from 6 to 48 h post injection (Supplementary Fig. 1). In contrast, there was no detectable RBD expression in placebo mice (Supplementary Fig. 1).

### Immunogenicity of a single mRNA-RBD vaccination.
The immune responses induced by the SARS-CoV-2 mRNA-RBD

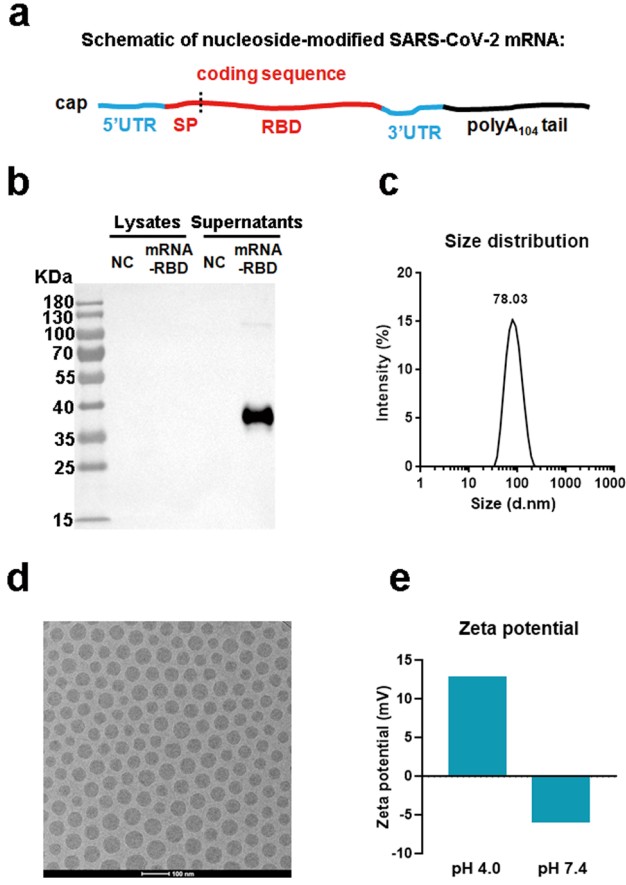

**Fig. 1 Construction and characterization of mRNA-RBD vaccine. a** Schematic of the mRNA-RBD vaccine design. The SARS-CoV-2 mRNA encodes the signal peptide (SP), receptor-binding domain (RBD) from SARS-CoV-2 strain Wuhan/IVDC-HB-01/2019. **b** mRNA-RBD was transfected into HEK293T cells. RBD expression in the cell lysate and supernatant was analyzed by western blotting. **c** Particle size of LNPs by dynamic light scattering. **d** A representative cryo-electron microscopy image of a LNPs solution following mRNA encapsulation. Scale bar, 100 nm. **e** Zeta potential for LNPs at pH 4.0 and 7.4. For **b** and **d**, two independent experiments were carried out with similar results. For **c** and **e**, one representative result from three independent experiments is shown. Source data are provided as a Source Data file.

vaccine candidate were analyzed in BALB/c mouse. Groups of mice ($n = 6$) were immunized i.m. once with a high dose (15 μg, mRNA-RBD-H) or low dose (2 μg, mRNA-RBD-L) or with a placebo. Serum samples obtained at 4 weeks after vaccination were evaluated for SARS-CoV-2 RBD-specific IgG and IgG subtypes by ELISA. As a result, the mean endpoint titers of RBD-specific IgG in mice immunized with the high dose rose to $>10^5$ and were significantly higher than those observed in mice immunized with the low dose (Fig. 2a). Additionally, both doses elicited IgG2a and IgG1 subclass RBD-specific antibodies, indicating a balanced Th1/Th2 response (Supplementary Fig. 2). Anti-SARS-CoV-2-neutralizing antibodies in serum of vaccinated mice were measured using two independent in vitro assays: pseudovirus neutralization assays and live SARS-CoV-2 neutralization assays. Pseudovirus neutralization assays showed that serum NT$_{90}$ (90% neutralization titers) in mice receiving the low dose (mean value 193) was significantly lower than that in mice receiving the high dose (mean value 727) (Fig. 2b). Live SARS-CoV-2 neutralization assays demonstrated that all of the vaccinated mice except one inoculated with the low dose developed

detectable neutralizing antibodies, and the mean NT$_{50}$ (50% neutralization titers) value in the high-dose group approached 920 and was 8.5- and 2.3-fold higher than those observed in the low-dose group and the human convalescent patient sera group, respectively (Fig. 2c). Therefore, we selected the high dose to immunize mice in the following studies.

Next, we investigated whether antibodies induced by SARS-CoV-2 mRNA-RBD could cross-react with or cross-neutralize the SARS-CoV and the MERS-CoV. ELISA-based binding assays demonstrated that vaccinated serum obtained from BALB/c mice at 4 weeks following immunization exhibited a strong binding capacity to SARS-CoV-2 RBD, whereas they showed substantially lower affinity to SARS-CoV RBD and little (if any) cross-reactivity to MERS-CoV RBD, respectively (Supplementary Fig. 3a–c). Pseudovirus cross-neutralization showed that no detectable cross-neutralization activities against either pseudo-typed SARS-CoV or MERS-CoV were observed, similar to the results of the placebo (Supplementary Fig. 3d). Therefore, these results indicated that although some cross-reactive antibodies elicited by SARS-CoV-2 mRNA-RBD exist, they may fail to block host cell entry mediated by the SARS-CoV-S protein.

To characterize the cellular immune responses induced by mRNA-RBD, enzyme-linked immunospot (ELISPOT) and intracellular cytokine staining (ICS) assays were performed. Spleens of C57BL/6 mice were harvested at 4 weeks post a single high-dose immunization. RBD-specific IgG and neutralizing antibodies were efficiently elicited in vaccinated mice (Fig. 2d, e). Splenocytes were isolated and evaluated for testing the capacity of secreting cytokines after re-stimulation with RBD peptide pools in vitro. ELISPOT assays demonstrated that T cells secreting gamma interferon (IFNγ) from mRNA-RBD-immunized mice were significantly more numerous than those from placebo-immunized mice (Fig. 2f). Moreover, immune responses mediated by SARS-CoV-2 RBD-specific CD4$^+$ and CD8$^+$ T cells were detected on the basis of intracellular IFNγ production by splenocytes of mRNA-RBD-immunized mice as evidenced in ICS assays (Fig. 2g, h). Therefore, mRNA-RBD can induce a substantial T cell response against SARS-CoV-2 RBD antigens aside from humoral immune responses.

It has been reported that the virus-binding but non-neutralizing antibodies induced by infection or vaccination may possibly elicit the ADE effect[46–50]. Thus, we also compared the differences in the proportions of neutralizing antibodies to binding antibodies between prime and boost immunization procedures. BALB/c mice ($n = 5$) were immunized with one (prime group) or two (boost group, 4-week internal) injections of mRNA-RBD at 15 μg or placebo via the i.m. route, and sera were collected at 8 weeks post initial injection for detection of RBD-specific and NAb titers. The results showed that in contrast to prime vaccination, boost injection improved binding antibody and NAb titers by 66- and 45-fold, respectively, and the ratios of NAb to binding antibody in the boost group were slightly lower (not significantly) than those in the prime group (Supplementary Fig. 4).

**Protection efficacy of mRNA-RBD in hACE2 transgenic mice against SARS-CoV-2.** To further explore the in vivo protection efficacy of mRNA-RBD against wild SARS-CoV-2 virus challenge, hACE2 transgenic mice ($n = 6$) received one (prime group) or two (boost group) immunizations of 15 μg mRNA-RBD or placebo via the i.m. route, and pre-challenge sera were collected for detection of specific and NAb titers (Fig. 3a). The results showed that a single immunization of mRNA-RBD induced robust production of RBD-specific IgG antibodies and neutralizing antibodies with a pseudovirus-neutralizing NT$_{90}$ titer of

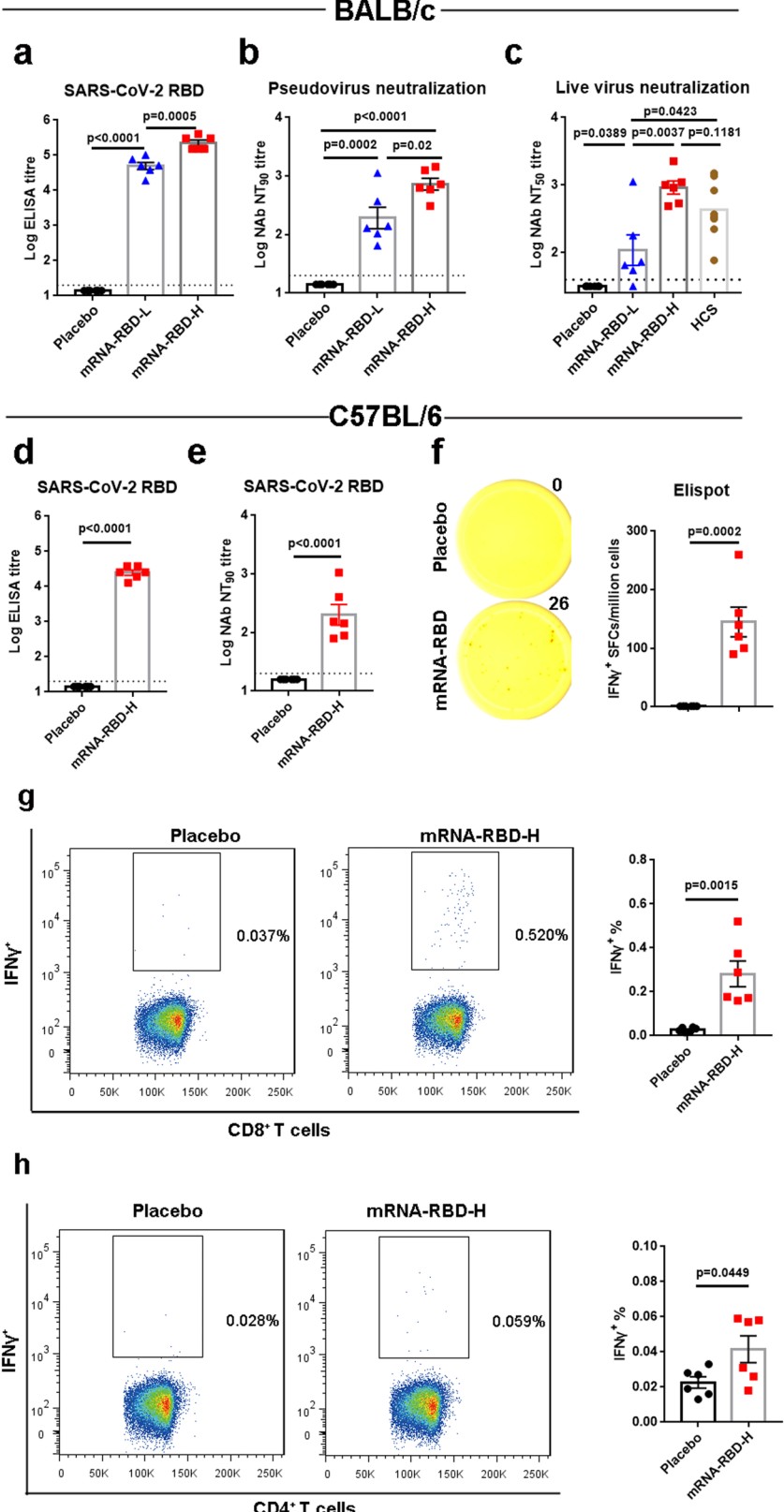

approximate 170 (Fig. 3b, c). In addition, a second immunization at 2 weeks post initial inoculation resulted in significant elevation of both IgG and neutralizing antibodies, with $NT_{90}$ titers reaching 2000, whereas no SARS-CoV-2-specific IgG or neutralizing antibodies were detected in sera from mice vaccinated with the placebo (Fig. 3b, c). Four weeks later, all mice were challenged with $1 \times 10^5$ FFU of SARS-CoV-2 via the intranasal (i.n.) route (Fig. 3a). We recorded the body weight for each mouse daily after infection for 5 days and found that the body weights of both prime and boost group mice showed a similar decrease at 1day post infection (dpi) but a faster increase during 3–5 dpi compared to the placebo group (Fig. 3d). Mice were euthanized at 5 dpi, and

**Fig. 2 Immunogenicity evaluation of a single mRNA-RBD vaccination. a–c** Groups of BALB/c mice ($n = 6$) were immunized with a single injection of mRNA-RBD at different doses or with a placebo via the i.m. route. Sera at 4 weeks post immunization were collected. SARS-CoV-2 RBD-specific IgG (**a**) and neutralizing antibody titers in sera against pseudovirus (**b**) and live virus (**c**) infection were determined. **d–h** C57BL/6 mice ($n = 6$) were inoculated with a single mRNA-RBD vaccination or a placebo. Serum samples were collected from mice at 4 weeks following vaccination. RBD-specific IgG titers and pseudovirus-neutralizing antibodies were measured as shown in **d** and **e**, respectively. **f** An ELISPOT assay was performed to evaluate the capacity of splenocytes to secrete IFNγ following re-stimulation with SARS-CoV-2 RBD peptide pools. **g**, **h** An ICS assay was conducted to quantify the proportions of IFNγ-secreting CD8+ (**g**) and CD4+ (**h**) T cells. mRNA-RBD-L indicates the low dose (2 μg). mRNA-RBD-H indicates the high dose (15 μg). HCS represents human convalescent sera. Data are means ± SEM (standard error of the mean). Comparisons were performed by Student's $t$-test (unpaired, two tailed). Placebo animals = black circles; mRNA-RBD-L vaccinated animals = blue triangles; mRNA-RBD-H vaccinated animals = red squares; HCS = brown circles; dotted line = the limit of detection. Data are one representative result of two independent experiments. Source data are provided as a Source Data file.

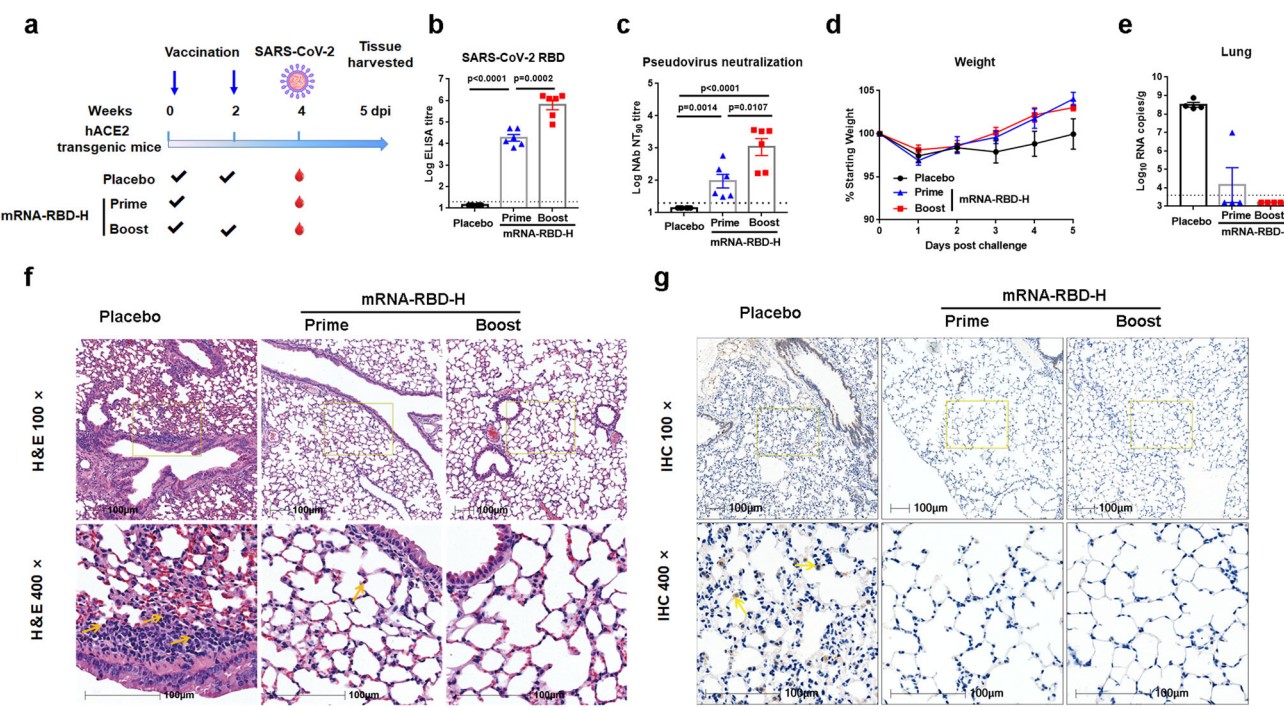

**Fig. 3 Protection efficacy of mRNA-RBD in hACE2 transgenic mice against SARS-CoV-2. a-d** Groups of hACE2 transgenic mice ($n = 6$) received one (prime group) or two (boost group) doses of mRNA-RBD-H or placebo via the i.m. route. Four weeks post initial vaccination, mice were challenged with $1 \times 10^5$ FFU of SARS-CoV-2 virus. **a** Mice immunization and challenge schedule. The blue arrows indicate the time of vaccination. **b**, **c** Sera collected at 4 weeks post initial vaccination were examined for IgG (**b**) and neutralizing antibody (**c**) titers. **d** Mice weight change after challenge. **e** Virus titers in lungs of challenged mice ($n = 4$). **f** Representative histopathology (H&E) of lungs in SARS-CoV-2-infected hACE2 mice (5 dpi). Infiltration of lymphocytes within alveolar spaces is indicated by yellow arrows. Scale bar, 100 μm. **g** Representative immunohistochemistry (IHC) of lung tissues with SARS-CoV-2 N-specific monoclonal antibodies. Virus is indicated by yellow arrows. Scale bar, 100 μm. mRNA-RBD-H indicates the high-dose vaccine (15 μg). Data are means ± SEM (standard error of the mean). Comparisons were performed by Student's $t$-test (unpaired, two tailed). Placebo animals = black circles; one injection-animals = blue triangles; two injections-vaccinated animals = red squares; dotted line = the limit of detection. Data are one representative result of two independent experiments. Source data are provided as a Source Data file.

harvested lungs were analyzed for viral RNA loads (four mice per group) and histopathological and immunohistochemical assays (two mice per group). The results for lung viral RNA loads demonstrated that all of the placebo-vaccinated mice became infected, with high levels of viral RNAs detected in the lungs ($\sim 10^{8.5}$ RNA copies equivalents per gram) (Fig. 3e). In contrast, mRNA-RBD-vaccinated mice were highly protected against SARS-CoV-2 infection (Fig. 3e). Seven out of eight mice—including three in the prime group and four in the boost group—had no detectable viral RNA in the lungs (Fig. 3e). A lower level of viral RNAs ($\sim 10^7$ RNA copies equivalents per gram) was detected in one mouse in the prime group, representing a 97% reduction in lung viral load compared to the control mice (Fig. 3e). This mouse exhibited the lowest $NT_{90}$ titer of 34 in the pseudovirus-neutralizing assay (Fig. 3c). The significance of a low-level lung viral load in one mouse, including implications for

a correlate of protection, warrants further study with a greater number of animals. We also evaluated whether antibody titer correlated with protection against virus challenge. The results showed an inverse correlation ($R^2 = 0.6976$, $P < 0.001$) between neutralizing antibody $NT_{90}$ titer and the level of viral RNA after SARS-CoV-2 challenge in hACE2 mice (Supplementary Fig. 5). Histopathological examination indicated that severe bronchopneumonia and interstitial pneumonia were observed in placebo mice, with edema and bronchial epithelia cell desquamation and infiltration of lymphocytes within alveolar spaces (Fig. 3f). In contrast, only very mild bronchopneumonia was observed in the prime group, and no lesions were observed in the boost group (Fig. 3f). Similarly, immunohistochemical assays detected SARS-CoV-2 viruses in the placebo mice, while no viruses were detected in the lungs of mRNA-RBD-vaccinated mice (Fig. 3g). These data demonstrated that a single vaccination of mRNA-RBD afforded

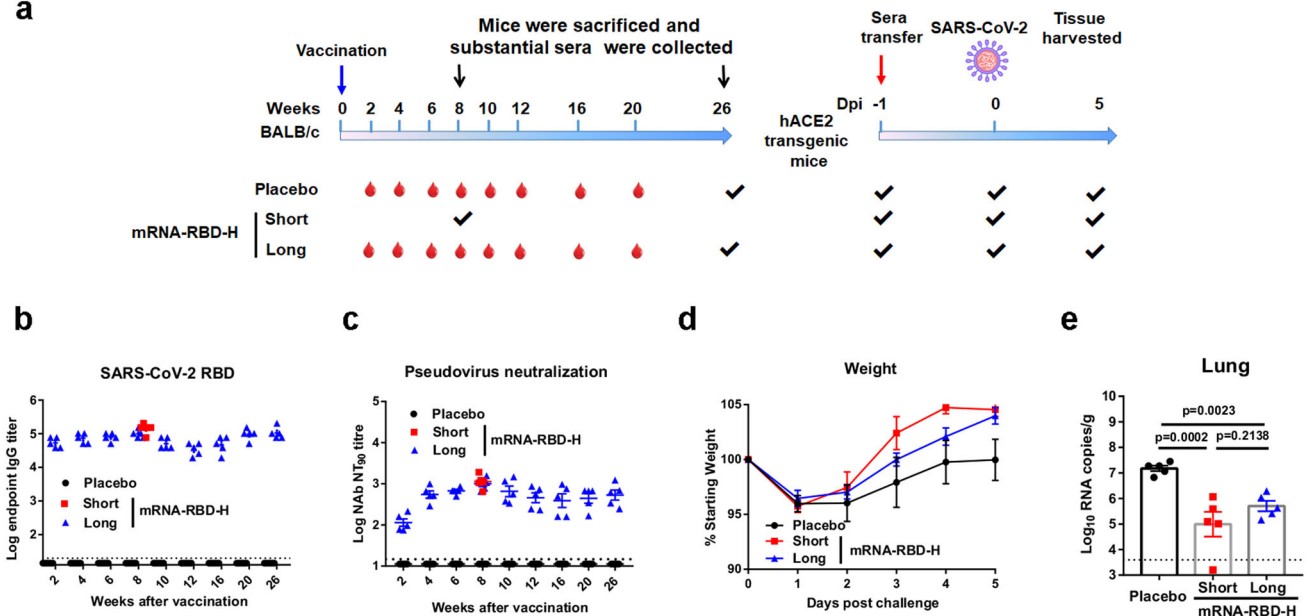

**Fig. 4 Duration and long-term protection of humoral response induced by mRNA-RBD. a** Passive immunization and challenge schedule. The blue and red arrow indicates the time of vaccination and sera transfer, respectively. **b, c** Groups of BALB/c mice (n = 10) received 15 µg of mRNA-RBD or a placebo. Half of the mice per group were euthanized at 8 weeks (short term) post vaccination, and massive sera were collected for further passive immunization. The other mice of the group were bled as desired and eventually euthanized at 26 weeks (long term) post vaccination to collect massive sera for further passive immunization. All serum samples were detected for IgG (**b**) and neutralizing antibodies (**c**) titers. **d–e** hACE2 transgenic mice (n = 5) were administered 350 µl per mouse of pooled short- and long-term immune sera and one day later were challenged with $1 \times 10^5$ FFU of SARS-CoV-2 via the i.n. route. **d** The hACE2 mice weight change was recorded after challenge. **e** Virus titers in lung. mRNA-RBD-H indicates the high-dose vaccine (15 µg). Data are means ± SEM (standard error of the mean). Comparisons were performed by Student's *t*-test (unpaired, two tailed). Placebo animals = black circles; animals for long-term study = blue triangles; animals for short-term study = red squares; dotted line = the limit of detection. Data are one representative result of two independent experiments. Source data are provided as a Source Data file.

remarkable prevention of SARS-CoV-2 replication in the lung and efficiently protected mice from lung lesions.

**Duration and long-term protection of humoral response induced by mRNA-RBD.** To evaluate the durability and long-term protection via the antibody response elicited by a single mRNA-RBD immunization, we performed an immunization and passive transfer study (Fig. 4a). BALB/c mice (n = 10) were immunized with mRNA-RBD-H or placebo via the i.m. route. Half of the mice per group (n = 5) were euthanized at 8 weeks post vaccination and massive sera were collected for further passive transfer as a short-term group. The remaining half of the mice per group (n = 5) were bled at designated times to explore the kinetics of induced humoral response, and the mice were finally euthanized at 26 weeks post vaccination, and massive sera were collected for further passive transfer as long-term group. The time-course study of RBD-specific IgG and pseudovirus-neutralizing antibodies demonstrated that both IgG and neutralizing NT90 titers in serum from vaccinated mice increased to peaks of ~$10^5$ and ~$10^3$, respectively, at week 8 (Fig. 4b, c). Noticeably, neutralizing NT90 titers were relatively stable during weeks 10–26 and remained at approximately 600 at 26 weeks post vaccination (Fig. 4c). Pooled immune or placebo sera (350 µl per mouse) were transferred to hACE2 mice (n = 5) by intraperitoneal injection (Fig. 4a). One day later, mice were infected with $1 \times 10^5$ FFU of SARS-CoV-2 via the i.n. route (Fig. 4a). Mouse body weight was monitored daily after challenge. The results showed that compared to placebo sera-transferred mice, both short-term and long-term sera-transferred mice displayed a similar weight loss during 1–2 dpi but then a rapid increase to a normal level from 3 to 5 dpi (Fig. 4d). Lung tissues were

harvested at 5 dpi and examined for viral RNA loads. As expected, lungs from animals treated with mRNA-RBD immune sera from the short-term and long-term group animals showed substantially reduced infectious virus burden (Fig. 4e). Consistent with the high neutralizing titers, a lower (but not significantly) viral RNA level was also observed in the lungs of animals receiving short-term sera compared to animals receiving long-term sera (Fig. 4e). The detection of viral RNA loads in hearts showed that only a very low level of virus RNA was detected in two mice from the placebo group, while no measurable viral RNAs were detected in all of the other mice (Supplementary Fig. 6). Together, these data suggest that a single immunization of mRNA-RBD conferred remarkably long-term protection against SARS-CoV-2 infection by eliciting a durable humoral response.

## Discussion
The ongoing pandemic of COVID-19 necessitates an urgent need for a safe and effective SARS-CoV-2 vaccine. Here, we developed a nucleoside-modified RBD-encoding mRNA vaccine against SARS-CoV-2 and demonstrated that a single immunization with mRNA-RBD induced robust NAb responses and provided near-complete protection against wild SARS-CoV-2 challenge in hACE2 transgenic mice. Noticeably, we also showed that a single dose of mRNA-RBD could induce a durable NAb response and thus confer a long-term protection in a sera passive transfer study. These data extend recent preclinical studies of inactivated virus, DNA, RNA, subunit, and adenovirus-vectored vaccines for SARS-CoV-2.

Since the beginning of the outbreak, the development of COVID-19 vaccines has proceeded at record speed. Peer-reviewed studies involving four SARS-CoV-2 RBD-based

mRNA vaccine candidates are currently available. Of those, two candidates (RBD-LNP and RBD mRNA-LNP) were mainly evaluated for immunogenicity in mice[15,35]; one (ARCoV) was investigated for immunogenicity in mice and non-human primates and for protective efficacy in a mouse model[21], and the last vaccine (BNT162b1) was evaluated for safety and immunogenicity in humans[34]. All of these RBD-based mRNA vaccine candidates have been demonstrated to possess excellent immunogenicity involving induction of potent neutralizing antibodies and cellular immune responses. For RBD mRNA-LNP, it has also been demonstrated that a single immunization elicited potent CD4[+] and CD8[+] T cell responses in lungs in addition to spleens, indicating a potential contribution to immune protection against SARS-CoV-2 infection[35]. Using a mouse-adapted strain of SARS-CoV-2, ARCoV showed marked immune protection in BALB/c mice with both one- and two-dose vaccination regimens[21]. Our study further demonstrated good immune protective efficacy of the RBD-based mRNA vaccine in vivo against the wild-type SARS-CoV-2 strain in hACE2 transgenic mice.

Although SARS-CoV-2 mRNA vaccine candidates in clinical trials have applied a two-dose regimen, a single nucleoside-modified mRNA vaccination has been shown to be sufficient to protect non-human primates against zika virus challenge[44]. Complete protection conferred by a single dose of Ad5-nCoV or Ad26-vectored vaccine against SARS-CoV-2 challenge in animal models has also currently been reported[51,52]. It has been shown that adenovirus-vectored ChAdOx1 nCoV-19 vaccine elicits specific IgG titer of 400-6400 and neutralizing titers of 5–40 (ref. [53]), whereas the DNA vaccine induced a NAb titer of 74–170 against live virus infection in immunized monkeys[12]. Here, we found that a single dose of mRNA-RBD elicited RBD-specific an IgG titer of $10^5$ and a live virus-NAb titer of 920. SARS-CoV-2 NAb geometric mean titer for human convalescent sera under our experimental platform was approximately 430. Consistent with the high IgG and NAb titers, a single dose of mRNA-RBD conferred near-complete protection against SARS-CoV-2 infection in vivo. In addition, the one-dose immunization procedure showed a similar proportion of NAb to binding antibody in immunized sera compared to the two-dose procedure, indicating that lack of a boost dose does not lead to an increased risk of ADE infection that is associated with virus-binding but non-neutralizing antibodies induced by vaccination. A single-shot SARS-CoV-2 vaccine would also have important logistic and practical advantages compared with a two-dose vaccine for mass vaccination campaigns and pandemic control.

Our study demonstrated that the high levels of NAb antibodies elicited by SARS-CoV-2 vaccine were maintained for at least 6.5 months, and probably much longer, since NAb titers were stable, and therefore conferred significant long-term protection against SARS-CoV-2 infection as evidenced in the sera passive transfer study. Consistent with the sustained NAb response in our study, the aforementioned RBD mRNA-LNP vaccine candidate has also demonstrated that a single immunization could induce potent long-lived plasma cells and memory B cell responses[35]. However, several previous studies reported that antibody-mediated immunity in convalescent humans from SARS-CoV-2 infection only persisted for 2–3 months[41,54]. One explanation for the difference in duration of humoral responses elicited by vaccination and infection may be that compared with the immune response to vaccination, the immune response to SARS-CoV-2 infection will target not only the RBD, the major target for potent neutralizing antibodies, but also non-RBD regions of the S and various other viral proteins that may readily induce a transient immune response. However, to elucidate the difference, further studies such as investigation of memory B cells and the antibody repertoire induced by vaccination and infection are obviously

warranted. Another limitation of our study is that we excluded the contributions of memory B cells and cellular immunity induced by vaccination in evaluation of long-term protection against SARS-CoV-2 by challenging sera-transferred rather than directly vaccinated mice, and thus further studies are also needed to address this question. Overall, our study has demonstrated near-complete protection via a single mRNA vaccination against SARS-CoV-2 challenge and provided valuable information concerning the durability of protective immune response induced by SARS-CoV-2 vaccines.

## Methods

**Ethics statement**. This study was carried out in strict accordance with the recommendations described in the Guide for the Care and Use of Laboratory Animals of the Institute of Microbiology, Chinese Academy of Sciences (IMCAS) Ethics Committee. All of the animal experiments were reviewed and approved by the Committee on the Ethics of Animal Experiments of IMCAS.

**Cells, viruses, and animals**. HEK293T cells (ATCC CRL-3216), Huh7 cells (3111C0001CCC000679), and Vero E6 cells (ATCC CRL-1586) were cultured at 37 °C in Dulbecco's modified Eagle's medium (DMEM) supplemented with 10% fetal bovine serum (FBS). SARS-CoV-2 virus hCoV-19/Wuhan/IVDC-HB-01/2019 (Accession ID: EPI_ISL_402119) used in this study was provided by Professor Wenjie Tan from the Chinese Center for Disease Control and Prevention. Vero E6 cells were applied to the amplification and titer titration of the virus stocks. BALB/c and C57BL/6 mice were purchased from Beijing Vital River Animal Technology Co., Ltd (licensed by Charles River) and were housed and bred in the temperature-, humidity- and light cycle-controlled animal facility (20 ± 2 °C; 50 ± 10%; light, 7:00–19:00; dark, 19:00–7:00) of specific-pathogen-free (SPF) mouse facilities in IMCAS. The female hACE2 transgenic mice were kindly provided by Professor Zhengli Shi from Wuhan Institute of Virology, CAS and were housed and bred in SPF mouse facilities in IMCAS. The animal experiments with SARS-CoV-2 challenge were conducted under animal biosafety level 3 (ABSL3) facility in IMCAS, which is approved for such use by of National Health Commission of China.

**mRNA production**. mRNA was produced using T7 RNA polymerase on linearized plasmids (synthesized by Genescript) encoding codon-optimized SARS-CoV-2 RBD glycoproteins (residues 319–541, accession number YP_009724390). The mRNA was transcribed to contain a 104 nucleotide-long poly(A) tail, and 1-methylpseudourine-5′-triphosphate was used instead of UTP to generate modified nucleoside-containing mRNA. The mRNA was purified by overnight LiCl precipitation at −20 °C, centrifuged at $18,800 \times g$ for 20 min at 4 °C to pellet, washed with 70% EtOH, centrifuged at $18,800 \times g$ for 1 min at 4 °C, and resuspended in RNase-free water. The purified mRNA was analyzed by agarose gel electrophoresis and stored frozen at −20 °C until use.

**mRNA transfection**. Transfection of HEK293T cells was performed with TransIT-mRNA (Mirus Bio) according to the manufacturer's instructions. In brief, mRNA (0.5 μg) was combined with TransIT-mRNA regent (1 μl) and boost reagent (1 μl) in 50 μl of serum-free medium, and the complex was added to $2.5 \times 10^5$ cells in 500 μl complete medium. The supernatant was collected and concentrated, and cells were lysed on ice in RIPA buffer (Beyotime) at 48 h after transfection.

**Western blot**. Whole-cell lysates and supernatants from cells transfected with mRNA-RBD were assayed for SARS-CoV-2 RBD expression by Western blotting. Samples were combined with loading buffer with dithiothreitol and separated by 12% SDS-PAGE. Transfer to a PVDF membrane was performed using a semi-dry apparatus (Ellard Instrumentation). The membrane was blocked with non-fat milk in TBS buffer containing 0.5% Tween-20. RBD protein was detected using serum from mice immunized with SARS-CoV-2 S1 proteins (Sino Biological) for 1 h, followed by secondary goat anti-mouse IgG-HRP (Santa Cruz) for 1 h. The membrane was developed by SuperSignal West Pic chemiluminescent substrate (Thermo Fisher Scientific).

**Lipid-nanoparticle encapsulation of mRNA**. SARS-CoV-2 RBD-encoded mRNA (mRNA-RBD) was encapsulated in LNPs using a self-assembly process in which an aqueous solution of mRNA at pH = 4.0 was rapidly mixed with a solution of lipids dissolved in ethanol. LNPs used in this study contained an ionizable cationic lipid, phosphatidylcholine, cholesterol, and PEG-lipid at a ratio of 50:10:38.5:1.5 mol/mol and were encapsulated at an mRNA to lipid ratio of around 0.05 (wt/wt). The mRNA-RBD LNPs were stored at 4 °C at a concentration of RNA of about 1 mg/ml.

**Particle size, zeta potential, and encapsulation efficiency**. Zetasizer was used to determine the particle size and zeta potential. Zeta potential was measured on

particles after suspending LNPs in deionized water at pH 4.0 and 7.4. The free and total mRNA concentrations in LNPs were determined using the Quant-iT Ribogreen RNA assay kit according to the manufacturer's instructions. The encapsulation efficiency (EE%) was calculated as follows: EE (%) = (1 − free mRNA concentration/total mRNA concentration) × 100.

**Cryo-electron microscopy of LNPs**. LNPs were transferred onto a glow-discharged ultrathin carbon-coated copper grid (Zhongjingkeyi Company) followed by 60 s of waiting and then blotted for 2 s with filter paper before plugging into liquid ethane using the Vitrobot Mark IV. The frozen grids were transferred at liquid nitrogen temperature and loaded into a Talos transmission electron microscope (Thermo Fisher Scientific) equipped with a field emission gun operated at 200 kV. The images were recorded on a direct electron detector (ED20) at a total electron dose of $\sim 50 e^-/\text{Å}^2$.

**Animal experiments**. For evaluation of RBD production from mRNA-RBD LNPs in vivo, female BALB/c mice aged 6–8 weeks ($n = 16$) were inoculated with 15 µg mRNA-RBD LNPs or poly(C) LNPs as a placebo control via the i.m. route. At 6, 12, 24, and 48 h post inoculation, four mice per group were sacrificed. Serum and tissues involving muscle, liver, kidney, spleen, and lung were collected immediately. Tissue samples were weighed and homogenized with a tissue grinder in 600 µl of DMEM. Samples were further centrifuged and the supernatant were stored at −80 °C until RBD quantification.

LNPs-encapsulated mRNA-RBD was diluted with PBS. Female BALB/c or C57BL/6 mice aged 6–8 weeks were inoculated with a high (15 µg) or low (2 µg) dose of mRNA-RBD LNPs or poly(C) LNPs via the i.m. route. Serum samples were collected at indicated times after vaccination, inactivated at 56 °C for 30 min, and stored at −20 °C.

For SARS-CoV-2 challenge experiments, hACE2 transgenic mice ($n = 6$) were immunized with one (prime group) or two (boost group) doses of mRNA-RBD via the i.m. route. Four weeks post initial immunization, mice were infected with $1 \times 10^5$ FFU of SARS-CoV-2 in a total volume of 50 µl of DMEM medium via the i.n. route. Animals were monitored daily for survival and weight loss. The mice were euthanized five days following challenge. Lung tissues were harvested for virus load detection (four mice per group) and pathological and immunohistochemical examination (two mice per group).

For passive immunization and SARS-CoV-2 challenge experiments, groups of BALB/c mice ($n = 10$) received 15 µg of mRNA-RBD or placebo. Half of the mice per group ($n = 5$) were euthanized at 8 weeks (short term) post vaccination. The other mice in the group ($n = 5$) were euthanized at 26 weeks (long term) post vaccination. The hACE2 transgenic mice ($n = 5$) were administered 350 µl per mouse of pooled immune sera collected from placebo or mRNA-RBD vaccinated mice via the intraperitoneal route. One day following serum transfer, mice were challenged with $1 \times 10^5$ FFU of SARS-CoV-2 via the i.n. route. Animals were monitored daily for survival and weight loss. Lung tissues were harvested for virus load detection at 5 days following challenge. All of the animal experiments with SARS-CoV-2 challenge were conducted under animal biosafety level 3 (ABSL3) facilities in IMCAS.

**Quantification of RBD expression in vivo**. Quantification of RBD expression in vivo was performed with a commercial SARS-CoV-2 RBD detection ELISA kit (Sino Biological) according to the manufacturer's instructions. The assay is based on a double-antibody sandwich principle that detects SARS-CoV-2 RBD protein in samples. Briefly, a monoclonal antibody specific for SARS-CoV-2 RBD protein was pre-coated onto plate wells. Standards, serum, and tissue homogenates were added to the wells and incubated for 2 h at room temperature. After three washes, plates were incubated with HRP-conjugated another anti-RBD antibody for 1 h at room temperature, followed by three washes and incubation with TMB substrate. The absorbance at 450 nm was measured. A standard curve of absorbance at 450 nm versus concentration was fit with a linear equation for accurate RBD quantification.

**Enzyme-linked immunosorbent assay**. ELISA plates (Corning) were coated overnight with 2 µg/ml of SARS-CoV-2 RBD, SARS-CoV RBD, or MERS-CoV RBD recombinant protein in 0.05 M carbonate-bicarbonate buffer, pH 9.6, and blocked in 5% skim milk in PBS. Serum samples were twofold serially diluted and added to each well. Plates were incubated with goat anti-mouse IgG-HRP, IgG1-HRP or IgG2a-HRP antibodies and developed with 3,3′,5,5′-tetramethylbenzidine (TMB) substrate. Reactions were stopped with 2 M hydrochloric acid, and the absorbance was measured at 450 nm using a microplate reader (PerkinElmer, USA). The endpoint titers were defined as the highest reciprocal dilution of serum to yield an absorbance greater than 2.1-fold of the background values. Antibody titer below the limit of detection was determined as half the limit of detection.

**Pseudovirus neutralization assay**. SARS-CoV-2, SARS-CoV, and MERS-CoV pseudovirus preparation and titration determination were performed as described previously[55]. Briefly, the plasmids of 14 µg pCAGGS-SARS-CoV-2-S, pCAGGS-SARS-CoV-S, or pCAGGS-MERS-CoV-S and 7 µg pNL4-3.luc.RE were cotransfected into HEK293T cells. After 48 h, the supernatant containing pseudovirus was harvested, centrifuged, and filtered through a 0.45-µm sterilized membrane. Single use aliquots were stored at −80 °C. The $TCID_{50}$ was determined by infection of

Huh7 cells. For the neutralization assay, 100 $TCID_{50}$/well was incubated with twofold serially diluted mouse sera for 30 min at 37 °C. The mixtures were then used to infect Huh7 cells seeded in 96-well plates. After 5 h incubation, the medium was replaced with DMEM containing 10% FBS, and the samples were incubated for an additional 24 h at 37 °C. Luciferase activity was measured using a GloMax 96 Microplate luminometer (Promega). The neutralization endpoint was defined as the fold-dilution of serum necessary for 90% inhibition of luciferase activity in comparison with virus control samples.

**Live SARS-CoV-2 neutralization assay**. The neutralizing activity of mice serum was assessed using a SARS-CoV-2 microneutralization assay. Briefly, heat-inactivated serum was twofold serially diluted and incubated with SARS-CoV-2 strain 01 (100 $TCID_{50}$) for 1 h at 37 °C. The mixture was added to pre-seeded Vero E6 cell monolayers in 96-well plates. After incubation for 48 h at 37 °C, the supernatant was collected for virus detection. The neutralization titers were defined as the serum dilution required for 50% neutralization of viral infection.

**Enzyme-linked immunospot (ELISPOT) assay**. To detect antigen-specific T lymphocyte responses, an IFN-γ-based ELISPOT assay was performed. Briefly, spleens were removed from vaccinated C57BL/6 mice at 4 weeks post immunization and splenocytes were isolated. Flat-bottom, 96-well plates were pre-coated with 10 g/ml anti-mouse IFNγ Ab (BD Biosciences, USA) overnight at 4 °C and then blocked for 2 h at RT. Mouse splenocytes were added to the plate ($1 \times 10^5$/well). Then, a peptide pool (2 µg/ml of individual peptide) consisting of 18-mers (overlapping by 10 amino acids) spanning the entire SARS-CoV-2 RBD protein was added to the wells. Phytohemagglutinin was added as a positive control. Cells incubated without stimulation were employed as a negative control. After 12 h of incubation, the cells were removed, and the plates were processed in turn with biotinylated IFNγ-detection antibody, streptavidin-HRP conjugate, and substrate. When the colored spots were intense enough to be visually observed, the development was stopped by thoroughly rinsing samples with deionized water. The numbers of the spots were determined using an automatic ELISPOT reader and image analysis software (Immuno Capture 6.5.0).

**ICS assay**. An ICS assay was performed to characterize antigen-specific CD4+ and CD8+ immune responses. Briefly, mouse splenocytes were added to the plate ($1 \times 10^6$ cells/well) and then stimulated with the peptide pool (2 µg/ml of individual peptide) for 4 h. The cells were incubated with Golgiplug (BD Biosciences) for an additional 12 h at 37 °C. Then, the cells were harvested and stained with anti-CD3, anti-CD4, and anti-CD8α surface markers (Biolegend). The cells were subsequently fixed and permeabilized in permeabilizing buffer (BD Biosciences) and stained with anti-IFN-γ (Biolegend). Flow cytometric analysis and cell sorting were performed on a BD FACSAria III flow cytometer (BD Biosciences) and the data were analyzed using FlowJo 7.6.1.

**Viral RNA extraction and RT-PCR**. Viral RNA was extracted from 100 µl of samples using the automated nucleic acid extraction system (TIANLONG, China) and following the manufacturer's instructions. Detection of the SARS-CoV-2 virus was performed using the One Step Prime Script RT-PCR kit (TaKaRa, Japan) on the Light Cycler 480 Real-Time PCR system (Roche, Rotkreuz, Switzerland) with primers. The following sequences were used: forward primer: 5′-AGAA-GATTGGTTAGATGATGATAGT-3′;
reverse primer:5′-TTCCATCTCTAATTGAGGTTGAACC-3′;
and probe:5′-FAM-TCCTCACTGCCGTCTTGTTG ACCA-BHQ1-3′.
Real-time RT-PCR was performed using the following conditions: 50 °C for 15 min and 95 °C for 3 min, 50 cycles of amplification at 95 °C for 10 s and 60 °C for 45 s.

**Histopathology and immunohistochemistry**. The lungs were fixed in 4% (v/v) paraformaldehyde solution for 72 h, and the paraffin sections (3–4 µm) were prepared routinely. Hematoxylin and eosin stain was used to identify histopathological changes in the lungs. The histopathology of the lung tissue was observed by light microscopy. For immunohistochemistry, SARS-CoV-2 N protein was detected using monoclonal antibody clone 019 (Sino Biological, China). Images were captured using a LEICA Versa 200 and were processed using software HALO v3.1.1076.379.

**Statistical analysis**. All of the data are expressed as the means ± standard errors of the means. For all of the analyses, P values were obtained from Student's t-test (unpaired, two tailed) or Spearman rank-correlation tests. All of the graphs were generated with GraphPad Prism version 7.0 software.

**Reporting summary**. Further information on research design is available in the Nature Research Reporting Summary linked to this article.

## Data availability

The authors declare that the data supporting the findings of this study are available within this paper or are available from the corresponding author upon reasonable request. Source data are provided with this paper.

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

## Acknowledgements

We are grateful to X. Qu and Q. Peng (Institute of Microbiology, CAS) for their help in performing cryo-electron microscopy of LNPs. This work is supported by the Strategic Priority Research Program of the Chinese Academy of Sciences (XDB29040201), the National Natural Science Foundation of China (NSFC) (81901680), National Key Research

and Development Project (2020YFC0842300), and the Technological Innovation Project of Shanxi Transformation and comprehensive Reform demonstration Zone (2017KJCX01).

## Author contributions

J.Y. and Q.H. designed the study; Q.H., K.J., S.T., F.W., B.H. conducted assays. Q.H., K.J., S.T., Z.T., W.T., J.H., and J.Y. analyzed and interpreted the data. Q.H. wrote the manuscript. Q.H., S.T., Q.W., G.F.G., and J.Y. discussed and edited manuscript.

## Competing interests

The authors declare no competing interests.
