## [Peer Review File · Nature Communications]

Reviewers' Comments:

Reviewer #1:

Remarks to the Author:
Comments

The examination of the lung viral titres was only performed on 4 out of the 6 mice challenged in each group (Figure 3e and main text). It is not possible to perform a statistical test on 4 animals and obtain a highly significant p value of <0.01. Can the authors remove that statistical inference from the figure.

Is there a reason why the single high dose vaccination (15 ug) was able to elicit a pseudovirus NT90 titre of 727 in the original dose finding experiment but only a 170 pseudovirus NT90 titre in the hACE2 transgenic mice? Please include the genetic background strain of the hACE2 mice.

A change from -5.7 mV at pH 7.4 to +12.9 mV at pH 4 is not 'intense' please remove the word intense.

The passive transfer experiments don't quite make sense to me. As the authors stated – why not directly vaccinated mice? Is this because the hACE2 mice failed to generate the same high levels of neutralizing antibody as the Balb/c mice? If so, the authors should reveal this caveat to their study – the natural vaccination would have been more interesting for duration of the total immune response (memory B cells, cellular) which the authors acknowledged. Also – if the hACE2 mice less good at producing neuts it's fine to say so, people are aware of the limitations of transgenic mice.

Minor Comments:

Figure 4a – please substitute the word 'substantial' for 'massive' in the figure (likewise – line 220).

Line 27 – replace 'antibodies' with 'antibody'

Lines 34 – 37 –

Replace 'Noticeably, high levels of neutralizing antibodies response induced by mRNA-RBD vaccination could maintain for at least 6.5 months and conferred a long-term remarkable protection for hACE2 transgenic mice against SARS-CoV-2 infection in sera transfer study.'

With 'Noticeably, the high levels of neutralizing antibodies induced by mRNA-RBD vaccination were maintained for at least 6.5 months and conferred a long-term notable protection for hACE2 transgenic mice against SARS-CoV-2 infection in a sera transfer study.'

Line 46 – replace 'leaded with 'led'

Line 60 -64 – This sentence doesn't make sense / a point

Line 92 – replace 'doses' with 'does'

Generally – the English is very good and I won't correct it in this review in each instance but the tenses, absence of a number of definite and indefinite articles and use of plurals in the text would benefit from modifications and correction by a native English speaker. This needs to be done for reading clarity.

Reviewer #2:

Remarks to the Author:

The manuscript evaluated the short- and long-term protective efficacy of a single dose of lipid-encapsulated RBD-encoding mRNA vaccine in hACE2 transgenic mice. This work is very valuable, but I still have the following confusion:

1. Line 208, author described the Fig. 3g as "...no viral in lung from all mRNA-RBD vaccinated mice". These is obvious positive signals representing the virus infection in prime group.
2. Fig. 3g demonstrated positive signal similar to boost group in prim group, but Fig. 3e show that the virus RNA detection in lung tissue of group B was almost negative.
3. It is recommended to design experiments to study the tissue distribution and duration of RBD expression in mice immunized with the mRNA-RBD vaccine. The RBD concentration in serum also should be measured.
4. Please analyze the correlation between neutralizing antibody titer and the level of viral RNA after SARS-CoV-2 challenge in hACE2 mice.
5. In sera transfer experiment (Fig. 4), whether the serum of hACE2 mice on day 0 (before challenge) was collected? It should provide the antibody titer data after transfer.
6. In Balb/C mice experiment (Fig. 2a), single dose of the vaccine could have elicited RBD-specific antibody titer exceed 5 Log, but the neutralizing antibody was slightly lower. Please compare the difference in the proportion of the neutralizing antibody in the binding antibody between prime and boost immunization procedures. And discuss the ADE risk and rationality of the single-dose vaccination procedure.
7. Due to the differences of the neutralization assay among different laboratories, the COVID-19 convalescent serum neutralizing antibody titer data under the experimental platform should be provided.

Reviewer #3:

Remarks to the Author:

Huang et al evaluated a nucleoside-modified SARS-CoV-2 vaccine in mice and found that it induced long-term protective responses after administration of a single vaccine dose.

The results have limited novelty as at least 3 other published papers have similar findings with RBD mRNA-LNPs (PMID: 32783919, PMID: 32795413, PMID: 32759966). The most valuable finding is the long-term protective efficacy data in hACE2 mice. The manuscript is clearly written. The reviewer has some minor claims, mainly clarifications.

Minor comments:

1. Please add the following paper to the introduction: PMID: 32783919
2. Indicate what mRNA-RBD-L and mRNA-RBD-H mean (I believe low and high but it is not explained in the text).
3. Indicate in the figure legends the number of independent experiments performed.
4. Please include mRNA-RBD-L data for B6 mice (Fig. 2 d-h) or explain why it was not added to the paper.
5. Lines 34-37. This is a bit misleading. Anti-RBD antibodies were produced in BALB/c mice and then transferred to hACE2 mice, which were challenged. This needs to be clarified because the reader has the impression that hACE2 mice were immunized initially and then challenged 6.5 months later.
6. Figure 4 legend: it is not clear if week 8 or week 26 serum was used for passive transfer shown in Fig. 4d-e.
7. In the discussion please discuss other published work that used RBD mRNA-LNP vaccines and highlight how the present study differs from published ones.
8. Check spelling throughout the paper (half mice per group, lines 216 and 635; ELISAPOT, line 612 etc...).
9. Line 312: how was the mRNA purified? Please provide details.

10. Line 434: RT-PCR: how do the authors know that they measured only live replicating virus and no dead challenge virus?

Response to reviewers' comments

We thank the three reviewers for their careful review of this manuscript. Each reviewer provided a number of constructive and thoughtful comments that were extremely helpful in revising this manuscript. We have now modified the manuscript according to their comments. Each point has been addressed.

The reviewers' comments are reproduced in their entirety in italics.

Reviewer #1 (Remarks to the Author):

Comments

(1) The examination of the lung viral titres was only performed on 4 out of the 6 mice challenged in each group (Figure 3e and main text). It is not possible to perform a statistical test on 4 animals and obtain a highly significant p value of <0.01. Can the authors remove that statistical inference from the figure.

Response: We agree with the reviewer that the number of mice in the lung viral examination is little small. The statistical inference has been removed. Please see Fig. 3e.

(2) Is there a reason why the single high dose vaccination (15 ug) was able to elicit a pseudovirus NT90 titre of 727 in the original dose finding experiment but only a 170 pseudovirus NT90 titre in the hACE2 transgenic mice? Please include the genetic background strain of the hACE2 mice.

Response: The hACE2 transgenic mice (HFH4-hACE2 mice) used in this study were from mixed genetic backgrounds, C3H and C57BL/6¹. As shown in Fig.2b and e, a single high dose elicited a pseudovirus NT90 titer of 727 in BALB/c mice, but only a titer of 200 in C57BL/6 mice. A published paper also reported similar results showing that the pseudovirus IC50 titers elicited by immunization twice with mRNA-1273 reached 819 in BALB/c but only 89 in C57BL/6². Thus, SARS-CoV-2 S or RBD antigens seem to be less immune response in C57BL/6 or hACE2 mice (C3H and C57BL/6 mixed background) compared with being in BALB/c mice.

1. Jiang, R.D. et al. Pathogenesis of SARS-CoV-2 in Transgenic Mice Expressing Human Angiotensin-Converting Enzyme 2. *Cell* **182**, 50-58.e58 (2020).
2. Corbett, K.S. et al. SARS-CoV-2 mRNA vaccine design enabled by prototype pathogen preparedness. *Nature* 586, 567-571 (2020).

(3) A change from -5.7 mV at pH 7.4 to +12.9 mV at pH 4 is not 'intense' please remove the word intense.

Response: We have deleted the word 'intense'. Please see lines 120.

(4) The passive transfer experiments don't quite make sense to me. As the authors stated – why not directly vaccinated mice? Is this because the hACE2 mice failed to generate the same high levels of neutralizing antibody as the Balb/c mice? If so, the authors should reveal this caveat to their study – the natural vaccination would have been more interesting for duration of the total immune response (memory B cells,

cellular) which the authors acknowledged. Also – if the hACE2 mice less good at producing neuts it's fine to say so, people are aware of the limitations of transgenic mice.

Response: The long-term protective efficacy experiment was initiated in early February 2020, and at that time hACE2 mice were not available. Based on a correlation of neutralizing antibody with protection as evidenced in SARS-CoV and MERS-CoV vaccine candidates, we changed to immunizing BALB/c mice instead of hACE2 mice to assess vaccine long-term protective efficacy by a passive transfer experiment. Of note, the hACE2 mice are indeed less good at producing neutralizing antibodies, in contrast to BALB/c mice as mentioned above.

Minor Comments:

(5) Figure 4a – please substitute the word ‘substantial’ for ‘massive’ in the figure (likewise – line 220).

Response: The word ‘substantial’ have been substituted for ‘massive’ in the figure. Please see lines 248 and 741.

(6) Line 27 – replace ‘antibodies’ with ‘antibody’

Response: The word has been substituted. Please see lines 27.

(7) Lines 34 – 37 –

Replace ‘Noticeably, high levels of neutralizing antibodies response induced by

mRNA-RBD vaccination could maintain for at least 6.5 months and conferred a long-term remarkable protection for hACE2 transgenic mice against SARS-CoV-2 infection in sera transfer study.’ With ‘Noticeably, the high levels of neutralizing antibodies induced by mRNA-RBD vaccination were maintained for at least 6.5 months and conferred a long-term notable protection for hACE2 transgenic mice against SARS-CoV-2 infection in a sera transfer study.’

Response: Thank you for the careful review. The sentence has been substituted. Please see line 34–37.

(8) Line 46 – replace ‘leaded with ‘led’

Response: The word has been replaced. Please see lines 46.

(9) Line 60 -64 – This sentence doesn’t make sense / a point

Response: Both SARS-CoV-2 RBD and full-length S are able to elicit potent neutralizing antibody responses. However, it has been reported that in contrast to RBD, the SARS-CoV full-length S protein contains several immunodominant epitopes that can induce non-neutralizing antibodies, including those associated with the ADE effect¹⁻⁴. A current study also demonstrated that SARS-CoV RBD could induce strong protection without obvious signs of the ADE effect, whereas full-length S protein induced weak protection and a strong ADE effect in a mouse model⁵. Thus, we deduce that compared with the full-length S protein, SARS-CoV-2 RBD may carry a lower risk of inducing an ADE effect by excluding certain epitopes in the S protein

that are prone to induce non-neutralizing and ADE-associated antibodies.

1. Tseng, C.T. et al. Immunization with SARS coronavirus vaccines leads to pulmonary immunopathology on challenge with the SARS virus. *PloS one* **7**, e35421 (2012).
2. Wang, Q. et al. Immunodominant SARS Coronavirus Epitopes in Humans Elicited both Enhancing and Neutralizing Effects on Infection in Non-human Primates. *ACS infectious diseases* **2**, 361-376 (2016).
3. Wang, S.F. et al. Antibody-dependent SARS coronavirus infection is mediated by antibodies against spike proteins. *Biochemical and biophysical research communications* **451**, 208-214 (2014).
4. He, Y. et al. Identification of immunodominant sites on the spike protein of severe acute respiratory syndrome (SARS) coronavirus: implication for developing SARS diagnostics and vaccines. *Journal of immunology (Baltimore, Md. : 1950)* **173**, 4050-4057 (2004).
5. Chen, W.H. et al. Yeast-expressed SARS-CoV recombinant receptor-binding domain (RBD219-N1) formulated with aluminum hydroxide induces protective immunity and reduces immune enhancement. *Vaccine* **38**, 7533-7541 (2020).

(10) Line 92 – replace ‘doses’ with ‘does’

Response: It has been replaced. Please see lines 92.

(11) Generally – the English is very good and I won't correct it in this review in each instance but the tenses, absence of a number of definite and indefinite articles and use of plurals in the text would benefit from modifications and correction by a native English speaker. This needs to be done for reading clarity.

Response: We feel so sorry for the mistakes in the manuscript and inconvenience they caused in your reading. The manuscript has been thoroughly revised and edited by a native English speaker. We hope it can meet the standard for final publication. Thank you very much for your careful review.

Reviewer #2 (Remarks to the Author):

The manuscript evaluated the short- and long-term protective efficacy of a single dose of lipid-encapsulated RBD-encoding mRNA vaccine in hACE2 transgenic mice. This work is very valuable, but I still have the following confusion:

(1) Line 208, author described the Fig. 3g as “...no viral in lung from all mRNA-RBD vaccinated mice”. There are obvious positive signals representing the virus infection

in prime group.

Response: Thank you for the careful review. According to Fig. 3g, the positive signals appears only in the bronchi, where virus was directly inoculated during SARS-CoV-2 challenge but not in lung alveolar cells in the prime group, whereas in the placebo group, there were very strong signals in the lung alveolar cells representing active virus replication apart from the bronchi. Moreover, massive infiltrations of lymphocytes within alveolar cavity could be observed in the placebo group, but not in the prime group. Thus vaccination with a single mRNA-RBD-H provided a remarkable protection. In our opinion, some positive signals in the bronchi in prime group mice may have resulted from residual challenge virus antigens or unwashed staining antibody in the experiment. To avoid misleading readers, we suggest replacing the original Fig. 3g with the revised one as followed below.

(2) Fig. 3g demonstrated positive signal similar to boost group in prim group, but Fig. 3e show that the virus RNA detection in lung tissue of group B was almost negative.

Response: As aforementioned, Fig. 3g (immunohistochemical assays) showed the appearance of possible residual challenged virus antigens in the bronchi of the prime group, whereas Fig. 3e (RT-PCR assays) showed almost none in the prime group in lung virus load detection. We think that the discordance may come from the fact that without active replication, input virus protein antigens should be more stable and sustain for a longer time compared with viral RNA. mRNA-1273 vaccine candidate has reported a similar individual case showing that lung of one vaccinated nonhuman primate was detected positive for virus antigen but negative for virus RNA¹. In order to avoid misleading readers, we suggest replacing the original Fig. 3g with the revised figure as well.

1. Corbett, K.S. et al. Evaluation of the mRNA-1273 Vaccine against SARS-CoV-2 in Nonhuman Primates. *The New England journal of medicine* (2020).

(3) It is recommended to design experiments to study the tissue distribution and duration of RBD expression in mice immunized with the mRNA-RBD vaccine. The RBD concentration in serum also should be measured.

Response: Thank you for the review. To examine the duration and distribution of RBD production from mRNA-RBD LNPs *in vivo*, groups of BALB/c mice (n=16) were immunized with 15 µg mRNA-RBD LNPs via the intramuscular route, and poly(C) RNA encapsulated in LNPs was used as a placebo control. Four mice per group were

ethanized at 6, 12, 24 and 48 h post injection, and serum, muscle, liver, kidney, spleen and lung samples were harvested. Tissue samples were weighed, homogenized with a tissue grinder in 600 μ l of DMEM, then centrifuged, and the supernatant were obtained. Serum and tissue homogenate samples were detected for RBD expression quantitatively by using a commercial SARS-CoV-2 RBD detection ELISA kit (SinoBiological). The results showed that in the mRNA-RBD group, the muscle and liver were the main RBD-expressing tissues, which is in accordance with previous observations for luciferase mRNA LNPs¹, and a very low level of RBD expression was also detected in serum and other tissue samples. Moreover, RBD expression in both muscle and liver tissues peaked at six hours post inoculation, with average concentrations of 553 and 437ng/ml and abated relatively quickly from 6 to 48 hours post injection. In contrast, there was no detectable RBD expression in placebo mice. The study has been added to the revised manuscript. Please see Extended Data Fig.1 and lines 124–137, 410–417, 442–452 and 755–762.

1. Pardi, N. et al. Expression kinetics of nucleoside-modified mRNA delivered in lipid nanoparticles to mice by various routes. *Journal of controlled release : official journal of the Controlled Release Society* **217**, 345-351 (2015).

(4) Please analyze the correlation between neutralizing antibody titer and the level of viral RNA after SARS-CoV-2 challenge in hACE2 mice.

Response: We analyzed the correlation between neutralizing antibody titer and the level of viral RNA after SARS-CoV-2 challenge in hACE2 mice. The results showed an inverse correlation ($R^2=0.6976$, $P<0.001$) between NT90 titers and the level of viral RNA in mouse lung. Please see Extended Data Fig. 5 and lines 229–232 and 795–800.

(5) In sera transfer experiment (Fig. 4), whether the serum of hACE2 mice on day 0 (before challenge) was collected? It should provide the antibody titer data after transfer.

Response: In the sera transfer experiment, immune sera were transferred to hACE2 mice by intraperitoneal injection and 24 hours later, mice were infected with

SARS-CoV-2 via the intranasal route. In order to maintain a good physiological status of hACE2 mice to be challenged, we did not bleed them.

(6) In Balb/C mice experiment (Fig. 2a), single dose of the vaccine could have elicited RBD-specific antibody titer exceed 5 Log, but the neutralizing antibody was slightly lower. Please compare the difference in the proportion of the neutralizing antibody in the binding antibody between prime and boost immunization procedures. And discuss the ADE risk and rationality of the single-dose vaccination procedure.

Response: Thank you for the review. BALB/c mice (n=5) were immunized with one (prime group) or two (boost group, 4-week interval) 15 µg doses of mRNA-RBD-H or with a placebo. Serum was collected eight weeks post initial immunization. The results demonstrated that the boost injection improved prime-induced binding and neutralizing antibody titers by 66- and 45-fold, respectively, and the ratios of neutralizing antibody to binding antibody in sera elicited by prime and boost immunization procedures were similar. Thus, lack of boost dose does not seem to lead to an increased risk of ADE infection, which is associated with virus-binding but non-neutralizing antibodies induced by vaccination. We have added these results and an ADE-related discussion to the revised manuscript. Please see Extended Data Fig. 4 and lines 187–198, 315–320, and 784–793.

(7) Due to the differences of the neutralization assay among different laboratories, the COVID-19 convalescent serum neutralizing antibody titer data under the experimental platform should be provided.

Response: The COVID-19 convalescent serum antibody titer data under our experimental platform have been provided. The results showed that SARS-CoV-2 neutralizing antibody geometric mean titer for a panel of eight human convalescent sera (HCS) was approximately 430. Please see Fig. 2c and lines 157–158, 312–313 and 716.

Reviewer #3 (Remarks to the Author):

Huang et al evaluated a nucleoside-modified SARS-CoV-2 vaccine in mice and found that it induced long-term protective responses after administration of a single vaccine dose.

The results have limited novelty as at least 3 other published papers have similar findings with RBD mRNA-LNPs (PMID: 32783919, PMID: 32795413, PMID: 32759966). The most valuable finding is the long-term protective efficacy data in hACE2 mice. The manuscript is clearly written. The reviewer has some minor claims, mainly clarifications.

Minor comments:

(1) Please add the following paper to the introduction: PMID: 32783919

Response: The paper (PMID: 32783919) was quoted in **Introduction** and **Discussion** sections. Please see lines 80–81, 287-289, 294-297, 312-313, and 619–621.

(2) Indicate what mRNA-RBD-L and mRNA-RBD-H mean (I believe low and high but it is not explained in the text).

Response: Yes, mRNA-RBD-L and mRNA-RBD-H indicates the low (2 µg) and high (15 µg) dose, respectively. We have added these explanations to the text. Please see lines 141–142.

(3) Indicate in the figure legends the number of independent experiments performed.

Response: We have indicated the number of independent experiments performed in the figure legends. Please see lines 698–701, 718–719, 734–735, 751–752, 760–761, 769–770, 781–782, 792, and 806-807.

(4) Please include mRNA-RBD-L data for B6 mice (Fig. 2 d-h) or explain why it was not added to the paper.

Response: We first evaluated immunogenicity of mRNA-RBD-L and mRNA-RBD-H in BALB/c mice, and found that mRNA-RBD-H elicited substantially more RBD-specific IgG and neutralizing antibody than mRNA-RBD-L as shown in Fig. 2 a-c and Extended Data Fig. 2. Thus, we selected mRNA-RBD-H to perform all of the subsequent studies involving evaluation of cellular immunity in C57BL/6 mice.

(5) Lines 34-37. This is a bit misleading. Anti-RBD antibodies were produced in BALB/c mice and then transferred to hACE2 mice, which were challenged. This needs to be clarified because the reader has the impression that hACE2 mice were immunized initially and then challenged 6.5 months later.

Response: We have replaced ‘Noticeably, high levels of neutralizing antibodies response induced by mRNA-RBD vaccination could maintain for at least 6.5 months and conferred a long-term remarkable protection for hACE2 transgenic mice against SARS-CoV-2 infection in sera transfer study.’ with ‘Noticeably, the high levels of neutralizing antibodies **in BALB/c mice** induced by mRNA-RBD vaccination were maintained for at least 6.5 months and conferred a long-term notable protection for

hACE2 transgenic mice against SARS-CoV-2 infection in a sera transfer study.’ The revised sentence highlights that it was the neutralizing antibody produced in BALB/c mice that conferred protection for hACE2 mice in the sera transfer study. Please see lines 34–37.

(6) Figure 4 legend: it is not clear if week 8 or week 26 serum was used for passive transfer shown in Fig. 4d-e.

Response: We have clarified in Figure 4 legend that both week 8 and week 26 serum were transferred to hACE2 mice for passive transfer study. Please see lines 746.

(7) In the discussion please discuss other published work that used RBD mRNA-LNP vaccines and highlight how the present study differs from published ones.

Response: There are currently four RBD-based mRNA vaccine candidates reported. Two candidates (RBD mRNA-LNP¹ and RBD-LNP²) were mainly investigated regarding immunogenicity in mice, while one (ARCoV)³ was evaluated for protective efficacy in an animal model, and the last vaccine (BNT162b1)⁴ was investigated regarding safety and immunogenicity in humans. All of these vaccine candidates showed excellent immunogenicity involving induction of potent neutralizing antibody and cellular immune responses. In particular, the study of RBD mRNA-LNP provided a very detailed immunogenicity evaluation and indicated that a single vaccination could elicit potent CD4⁺ and CD8⁺ T cell responses in lungs in addition to the spleens, possibly contributing to immune protection against SARS-CoV-2 infection¹. Using a

mouse-adapted strain of SARS-CoV-2, ARCoV has been reported that both one- and two-dose vaccination regimens afforded marked immune protection to BALB/c mice³. Our study further demonstrated the protective efficacy *in vivo* of RBD-based mRNA vaccine against wild-type SARS-CoV-2 strain in hACE2 transgenic mice. More importantly, our study first indicated that vaccination-induced neutralizing antibodies were maintained for at least 6.5 months and afforded long-term notable protection against SARS-CoV-2 in the sera transfer study. These discussion contents have been added to the revised manuscript. Please see lines 285–301 and 323–327.

1. Laczkó, D. et al. A Single Immunization with Nucleoside-Modified mRNA Vaccines Elicits Strong Cellular and Humoral Immune Responses against SARS-CoV-2 in Mice. *Immunity* (2020).
2. Tai, W. et al. A novel receptor-binding domain (RBD)-based mRNA vaccine against SARS-CoV-2. *Cell research*, 1-4 (2020).
3. Zhang, N.N. et al. A Thermostable mRNA Vaccine against COVID-19. *Cell* (2020).
4. Mulligan, M.J. et al. Phase 1/2 study of COVID-19 RNA vaccine BNT162b1 in adults. *Nature* (2020).

(8) Check spelling throughout the paper (half mice per group, lines 216 and 635; ELISAPOT, line 612 etc...).

Response: We feel so sorry for the spelling mistakes in the manuscript and inconvenience they caused in your reading. The manuscript has been thoroughly checked and revised.

(9) Line 312: how was the mRNA purified? Please provide details.

Response: mRNA was purified by overnight LiCl precipitation at -20°C, centrifuged at 14,000 RPM for 20 min at 4°C to pellet, washed with 70% EtOH, centrifuged at 14,000 RPM for 1 min at 4°C and resuspended in RNase-free water. The purification method has been added to **Methods**. Please see lines 368–371.

(10) Line 434: RT-PCR: how do the authors know that they measured only live replicating virus and no dead challenge virus?

Response: We agree with the reviewer that RT-PCR assay indeed has a limitation in that it cannot distinguish replicating virus from dead challenge virus. However, RT-PCR also takes advantages of rapidity, sensitivity and reproducibility, and various SARS-CoV-2 vaccine candidates have used RT-PCR to determine the viral titers in animal lungs after challenge¹⁻⁵. Thus, we still chose RT-PCR to measure lung viral titers.

1. Zhang, N.N. et al. A Thermostable mRNA Vaccine against COVID-19. *Cell* (2020).
2. Wang, H. et al. Development of an Inactivated Vaccine Candidate,

BBIBP-CorV, with Potent Protection against SARS-CoV-2. *Cell* **182**, 713-721.e719 (2020).

3. Gao, Q. et al. Rapid development of an inactivated vaccine candidate for SARS-CoV-2. *Science*, eabc1932 (2020).
4. Wu, Y. et al. A noncompeting pair of human neutralizing antibodies block COVID-19 virus binding to its receptor ACE2. *Science* (2020).
5. Mercado, N.B. et al. Single-shot Ad26 vaccine protects against SARS-CoV-2 in rhesus macaques. *Nature* (2020).

Reviewers' Comments:

Reviewer #1:

Remarks to the Author:

The authors have addressed my comments from the initial review. The revisions and the revisions in response to the other two reviewers have improved the paper.

Reviewer #2:

Remarks to the Author:

The revised submission has been greatly improved and addressed all of my comments.